# CORTEX: CONCEPT-ORIENTED TOKEN EXPLANATION IN VECTOR-QUANTIZED GENERATIVE MODEL

## ABSTRACT

Vector-Quantized Generative Models (VQGMs) have emerged as powerful tools for image generation. However, the key component of VQGMs—the codebook of discrete tokens—is still not well understood, e.g., which tokens are critical to generate an image of a certain concept? This paper introduces Concept-Oriented Token Explanation (CORTEX), a novel approach for interpreting VQGMs by identifying concept-specific token combinations. Our framework employs two methods: (1) a saliency-based method that analyzes token saliency value in individual images, and (2) an optimization-based method that explores the entire codebook to find globally relevant tokens. Experimental results demonstrate COR-TEX's efficacy in providing clear explanations of token usage in the generative process, outperforming baselines across multiple pretrained VQGMs. CORTEX not only improves VQGM transparency but also enables tasks such as targeted image editing, offering valuable insights into the model's internal representations.

## 1 INTRODUCTION

Vector-Quantized Image Generative Models (VQGMs) (Ramesh et al., 2021; Esser et al., 2021; Yu et al., 2021; Jin et al., 2023) have become powerful tools for high-quality image generation using discrete latent space representations. A critical component of these models is the *codebook* (Esser et al., 2021), which acts as a learned dictionary of visual elements. This codebook stores a finite set of discrete tokens, each representing various patterns or features within an image, such as textures, shapes, or object parts. During the generation process, the model selects tokens from this codebook to compose the final image. However, these high-dimensional tokens are difficult to interpret, making it challenging to understand how specific concepts are represented in the generative process. Moreover, not all tokens contribute equally to the generation of a particular concept (e.g., object categories or visual attributes), leading to the need for methods that can distinguish between concept-relevant and background tokens. Improving the interpretability of these tokens can provide deeper insights into how VQGMs represent semantic concepts, enabling more precise control over the generation process and facilitating tasks such as targeted image editing.

A straightforward approach to token interpretation may involve selecting tokens frequently appearing in images generated for a specific concept/object (Blei et al., 2003). However, this naive method often selects tokens that represent *contextual or background* elements, resulting in explanations cluttered with irrelevant information. This inability to differentiate between essential and non-essential tokens hinders clear understanding of how the model represents specific concepts.

To address this issue, we draw on the *Information Bottleneck* principle (Tishby et al., 2000), which focuses on compressing input data while retaining the most relevant information for a given task. In the context of VQGMs, we apply this principle to identify and preserve the tokens that are most informative for a specific concept, while filtering out those that contribute little to its representation.

In this paper, we propose **CORTEX** (Concept-Oriented Token Explanation), a novel framework that interprets VQGMs by identifying concept-specific token combinations. CORTEX comprises two methods: a saliency-based method that analyzes individual token importance for a given image and an optimization-based method that explores the entire codebook to find globally relevant tokens for a concept. By focusing on the most critical tokens and filtering out non-essential information, COR-

TEX provides clearer, more interpretable explanations of how VQGMs generate specific concepts. This, in turn, enables precise image manipulation and editing based on token-level representations.

Experimental results demonstrate the effectiveness of CORTEX according to the pretrained VQGM, including ResNet and Vision Transformer architectures. Our saliency-based method consistently identifies the most relevant tokens for concept-specific image generation, while the optimization-based method allows targeted image editing by manipulating these tokens. Together, these methods significantly enhance the transparency and controllability of VQGMs, providing valuable insights into the model's internal representations and offering practical tools for downstream generative tasks. Our main contributions are summarized as follows:

- We propose a **saliency-based method** for input-dependent token explanation, which identifies concept-relevant tokens in generated images by leveraging an information extractor based on the Information Bottleneck principle.
- We introduce an **optimization-based method** that explores the entire codebook to find globally relevant tokens for concept explanation, utilizing the same information extractor to optimize token combinations independent of specific inputs.
- Our experiments validate both **token-based explanation methods**, demonstrating their effectiveness in enhancing VQGM interpretability and enabling applications such as precise image editing.

## 2 PRELIMINARY

### 2.1 VECTOR QUANTIZED IMAGE GENERATIVE MODEL

Vector Quantized Image Generative Models (VQGM) (Ramesh et al., 2021; Esser et al., 2021; Yu et al., 2021; Jin et al., 2023) generate images via decoding from discrete tokens. These models are typically designed for conditional generation and are capable of creating images based on given concepts, such as text descriptions. During the image generation process, these models have three key parts: a codebook that stores token information, a transformer that predicts tokens based on the codebook and the concepts, and a decoder that turns tokens into images. It is worth noting that the term "concept" refers to the "input condition" in conditional generation, which guides the image generation process. The codebook plays a crucial role in VQGM by quantizing continuous high-dimensional visual features into discrete tokens. It is obtained through training on a large volume of image data to encode a diverse set of visual elements. However, tokens in the codebook are difficult to interpret directly because they represent high-dimensional features extracted from image pixels, capturing complex visual patterns that are not intuitively comprehensible to humans.

Let $G$ be a VQGM model with a codebook $\mathcal{C} \in \mathbb{R}^{K \times d} = [t_0, \ldots, t_{K-1}]^\top$, where $K$ is the total number of unique tokens, and $t_i \in \mathbb{R}^d$ is a $d$-dimensional vector representing the token $i$. The codebook $\mathcal{C}$ can be viewed as a look-up table. To generate an image, the model first uses its transformer to predict a sequence of $m^2$ tokens according to the input concept. It then looks up each token's corresponding vector in the codebook. These vectors are arranged into a 3D tensor $\mathbf{E} \in \mathbb{R}^{d \times m \times m}$, which we call the token-based embedding. Finally, the decoder transforms this embedding $\mathbf{E}$ into an $H \times H$ image, where $H$ is typically larger than $m$. This $\mathbf{E}$ is central to our analysis, as it directly shows how the model uses tokens to create images.

### 2.2 INFORMATION BOTTLENECK

The Information Bottleneck (IB) principle, introduced by (Tishby et al., 2000), provides a theoretical framework for extracting relevant information from data. The core idea of IB is to compress an input variable $X$ into a representation $T$ that retains maximal information about a target variable $Y$. Formally, this is achieved by minimizing the following objective:

$$\mathcal{L}_{\text{IB}} = I(X; T) - \beta I(T; Y), \tag{1}$$

where $I(\cdot; \cdot)$ denotes mutual information, and $\beta > 0$ is a hyperparameter that controls the trade-off between compression (minimizing $I(X; T)$) and preservation of relevant information (maximizing $I(T; Y)$). This principle has found applications in various machine learning tasks (Tishby & Zaslavsky, 2015; Hafez-Kolahi & Kasaei, 2019; Goldfeld & Polyanskiy, 2020), particularly in feature

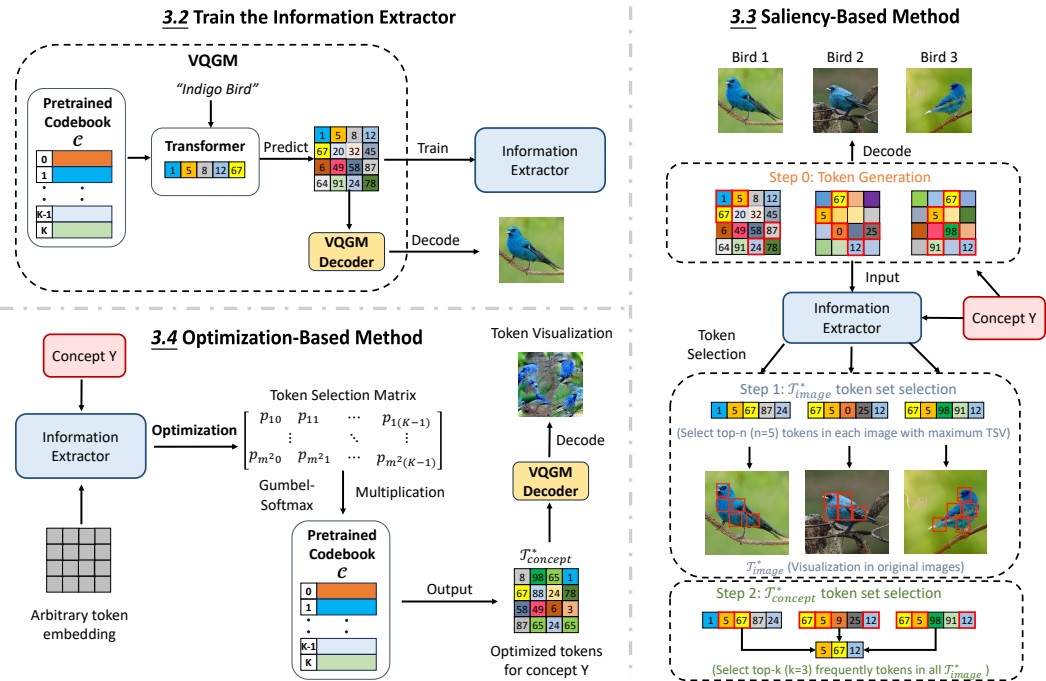

Figure 1: CORTEX comprises three key components: an Information Extractor and two explanation methods. The saliency-based method identifies concept-relevant tokens for each individual image, while the optimization-based method generates token combinations that represent the concept.

selection and representation learning, where it helps identify the most relevant features for a given concept while discarding irrelevant or redundant information. In the context of our work, the IB principle inspires our approach to extracting concept-specific tokens from VQGMs, focusing on the most relevant tokens while filtering out unnecessary background tokens.

# 3 METHODOLOGY

## 3.1 OVERVIEW

We propose a novel framework for concept-specific token explanation in VQGMs, aiming to identify the most relevant tokens within the discrete token space for a user-specified concept as the token-based explanation. Here, a concept can refer to a certain aspect of image content, such as object categories (e.g., "cat" or "dog"), visual attributes (e.g., "red" or "shiny"), or more abstract concepts (e.g., "rural scene" or "urban scene"). Our approach is inspired by the Information Bottleneck principle, focusing on extracting the most pertinent tokens to the concept while filtering out background information. The core of our framework is an **information extractor** introduced in Section 3.2 that predicts the concept of interest from token-based embedding $\mathbf{E}$. It is important to note that since our goal is to explain the behavior of VQGMs, all token-based embeddings $\mathbf{E}$ used in this study are generated by the VQGM itself, rather than derived from real images.

Recognizing that users are often interested in explanations for images generated during inference, we first introduce an **input-dependent** method based on saliency analysis in Section 3.3 to identify tokens that are most relevant to the concept in a specific image. Furthermore, we extend this approach to a concept-level analysis by aggregating identified tokens across a diverse set of images associated with the concept $Y$. This aggregation yields a more comprehensive, yet still input-dependent, view of how the model represents the concept across various instances. To complement the input-dependent method, we also present an **input-independent** method in Section 3.4. This approach identifies globally relevant token combinations for the concept by exploring the entire codebook, rather than relying on sampled images. Through an optimization process, we select tokens that best explain the user-specified concept $Y$, providing a comprehensive token-based explanation that considers all possible combinations within the codebook.

Both methods ultimately aim to obtain a token combination $\mathcal{T}^*$ which is the token-based explanation of the concept $Y$. It represents a subset of tokens from the VQGM's codebook that effectively characterize or explain the given concept $Y$. This combination of tokens captures the essential visual or semantic features that are associated with the concept to be explained.

## 3.2 INFORMATION EXTRACTOR

The Information Extractor serves as the cornerstone of our framework, designed to identify the most relevant tokens for a given concept. This model operates on token-based embeddings $\mathbf{E} \in \mathbb{R}^{d \times m \times m}$, generating predictions related to the concept of interest $Y$:

$$y = f_Y(\mathbf{E}), \tag{2}$$

where $f_Y$ is a classification model specifically trained for predicting concept $Y$. It determines whether the token-based embedding $\mathbf{E}$ contains the concept $Y$. For instance, if $Y$ denotes a particular image label, then $f_Y$ is a classifier capable of identifying the presence of that label in the decoded image based on $\mathbf{E}$. The output $y$ represents the model's prediction, indicating the likelihood that $\mathbf{E}$ contains concept $Y$. The model's architecture and training objective align closely with the Information Bottleneck principle (Saxe et al., 2019; Shwartz-Ziv & Tishby, 2017). To accurately predict or characterize the concept $Y$, the model must inherently focus on tokens and token combinations that are most representative and relevant to $Y$, while minimizing their reliance on irrelevant input tokens. This behavior mirrors the core tenets of the Information Bottleneck framework: maximizing information about the target variable while compressing the input.

Given an information extractor $f_Y$ and a target concept $Y$, we aim to find the optimal token combination $\mathcal{T}^*$ that servers as the explanation of concept $Y$:

$$\mathcal{T}^* = \phi(f_Y, [\mathbf{E}]), \tag{3}$$

where the function $\phi$ selects the most relevant tokens based on the trained information extractor $f_Y$, the concept $Y$ that needs to be explained, and optionally the input embeddings $\mathbf{E}$. Specifically, $\phi$ encompasses both our saliency-based and optimization-based approaches for token selection.

## 3.3 SALIENCY-BASED TOKEN EXPLANATION

As users are often interested in explanations of existing generated images, we first propose a **input-dependent** method to identify the most relevant tokens in these images. This method uses saliency analysis (Simonyan, 2013) to determine how each token in the input embedding $\mathbf{E}$ contributes to the prediction of the concept $Y$. This approach allows us to pinpoint which specific tokens are most crucial for representing the given concept in the context of a particular input image, providing a fine-grained, instance-specific explanation of the model's behavior.

Given an input token-based embedding $\mathbf{E} \in \mathbb{R}^{d \times m \times m}$, we compute the saliency score of each token using the following equation:

$$\bar{\nabla}\mathbf{E} = \frac{1}{N} \sum_{l=1}^{N} \nabla_{\mathbf{E}} f_Y(\mathbf{E} + \boldsymbol{\epsilon}_l), \tag{4}$$

where $f_Y(\mathbf{E})$ is the output of our information extractor for the concept $Y$, $N$ is the number of samples, and $\boldsymbol{\epsilon}_l \sim \mathcal{N}(0, \sigma^2 \mathbf{I})$ with $\sigma = \alpha(\max(\mathbf{E}) - \min(\mathbf{E}))$. The resulting $\bar{\nabla}\mathbf{E} \in \mathbb{R}^{d \times m \times m}$ has the same dimensions as $\mathbf{E}$. We then calculate the Token Saliency Value (TSV) for each token $t_i$ in the $m \times m$ grid. The TSV serves as a measure of the importance or relevance of the token to the prediction of the concept $Y$, with higher values indicating stronger associations between the token and the concept. $\text{TSV}(t_i)$ is computed by taking the maximum value across all $d$ channels of the gradient at the specific position corresponding to the token $t_i$ as follows:

$$\text{TSV}(t_i) = \max_{1 \leq j \leq d} |\bar{\nabla}\mathbf{E}(j, p_i)|, \tag{5}$$

where $p_i$ represents the position of token $t_i$ in the $m \times m$ grid, and $\bar{\nabla}\mathbf{E}(p_i, j)$ denotes the gradient value at position $p_i$ in the $j$-th channel. This operation reduces the $d$-dimensional gradient vector

at each token's position to a scalar value, representing the token's relevance to concept $Y$. After calculating the TSVs, we identify relevant token combinations at two levels:

$$\mathcal{T}^*_{\text{image}} = \text{Top}n(t_i : i \in 1, \ldots, m^2, \text{key} = \text{TSV}),$$

$$\mathcal{T}^*_{\text{concept}} = \text{Top}k(\bigcup_{\text{sampled images}} \mathcal{T}^*_{\text{image}}, \text{key} = \text{Freq}). \quad (6)$$

Here, $\mathcal{T}^*_{\text{image}}$ represents the Top $n$ tokens with the highest TSV for each specific image, providing an image-specific explanation. $\mathcal{T}^*_{\text{concept}}$ aggregates these image-specific sets across all sampled images related to the concept and selects the $k$ most frequent tokens, offering a concept-level explanation for $Y$. It is important to note that while $\mathcal{T}^*_{\text{concept}}$ provides a broader perspective, it is still input-dependent, as it is derived from all generative images related to the concept $Y$. This dual-level approach allows us to capture both instance-specific patterns (through $\mathcal{T}^*_{\text{image}}$) and concept-level trends (through $\mathcal{T}^*_{\text{concept}}$) in how the model represents and utilizes key information for the concept $Y$.

### 3.4 OPTIMIZATION-BASED TOKEN EXPLANATION

The saliency-based method analyzes tokens in individual-generated images but may not effectively explore the entire token space. To address this limitation, we propose an optimization-based, input-independent approach to identify the most relevant token combinations that explain a specific concept $Y$. Our method offers a global perspective by exploring all possible token combinations in the codebook, relying solely on the information extractor $f_Y$ rather than any specific input. Since we optimize over the input, it can be arbitrary—such as random noise or any existing token-based embedding. Building on previous work that uses input optimization for model interpretation (Nguyen et al., 2016; Erhan et al., 2009; Yosinski et al., 2015), our approach uniquely focuses on optimizing the token selection matrix instead of operating in the pixel space.

Given an information extractor $f_Y$ and a target concept $Y$, we aim to find the optimal token combination $\mathcal{T}^*$ that maximizes the model's prediction for $Y$, as defined in Equation 3. This method is independent of any specific input embedding, allowing us to explore the entire token space defined by the codebook. We define a **token selection matrix** $P \in \mathbb{R}^{m^2 \times K}$, where $m^2$ is the total number of token positions in the embedding $\mathbf{E}$, and $K$ is the size of the codebook. Each row of $P$ corresponds to a token position in $\mathbf{E}$ and contains a probability distribution over the $K$ possible tokens. We apply a binary mask $M_{\text{mask}} \in \{0, 1\}^{m^2}$, where $M_{\text{mask}}$ is a one-dimensional vector of size $m^2$. Each element in $M_{\text{mask}}$ corresponds to a row in $P$, with 1 indicating optimization for this token and 0 indicating a fixed position (keeping the original token unchanged). This selective optimization allows us to target relevant positions in the token selection process. We employ the Gumbel-Softmax technique (Jang et al., 2016) for differentiable token selection, as directly selecting discrete tokens from the codebook is not differentiable. This technique transforms the discrete selection process into a continuous, differentiable operation, enabling the use of gradient-based optimization algorithms to find the optimal token combinations.

$$E = \text{GumbelSoftmax}(P, \tau) \times C, \quad (7)$$

where $\text{GumbelSoftmax}(P, \tau) \in \{0, 1\}^{m^2 \times K}$ is a one-hot matrix representing the selected tokens, $\tau$ is the temperature parameter, and $C \in \mathbb{R}^{K \times d}$ is the codebook matrix (details can be found in Appendix A.2). The optimization of $P$ is conducted using:

$$P_{k+1} = P_k - \eta(\nabla_P \mathcal{L}(P_k) \odot M^T_{\text{mask}}),$$

$$\mathcal{L}(P) = -f_Y(E) + \alpha \|E\|_2^2, \quad (8)$$

where $\alpha$ is a regularization parameter and $\eta$ is the learning rate. After the optimization process converges, we obtain the final selection matrix $P^*$. The optimal token combination $\mathcal{T}^*$ is then derived from $P^*$ for the unmasked positions:

$$\mathcal{T}^*_{concept} = \{t_k : k = \arg\max_j P^*_{i,j}, \forall i \text{ where } M_{\text{mask},i} = 1\}. \quad (9)$$

This $\mathcal{T}^*_{concept}$ represents the set of tokens in the unmasked positions, which best explains the target concept $Y$ according to our information extractor $f_Y$. This method provides a unique perspective by identifying the most relevant tokens for explaining concept $Y$ across all possible token combinations, rather than being constrained to tokens from specific generated images. It reveals how the model globally represents the concept, offering insights into the fundamental token combinations that best characterize $Y$ in the context of VQGMs.

Table 1: Comparison of Information Extractor prediction accuracy (%).

| Model | Top 1 | Top 3 | Top 5 | Top 10 |
|---|---|---|---|---|
| CNN-based Extractor | 53.07 | 71.37 | 77.73 | 84.65 |
| ResNet-based Extractor | 51.43 | 69.23 | 76.00 | 83.12 |

## 4 EXPERIMENTS

### 4.1 EXPERIMENTAL SETUP

Our proposed framework aims to explain concept-specific information in VQGMs over a diverse range of concepts. Consistent with prior research (Chefer et al., 2021; Binder et al., 2016; Simonyan, 2013) we adopt the methodology of treating each category in the ImageNet (Deng et al., 2009) challenge dataset as a distinct concept to be explained. The experiments are designed to verify that the token combinations selected by CORTEX are indeed the most relevant to the concept (label) being explained. In this experiment, the information extractor $f_Y$ is trained as an image classifier with 1,000 ImageNet categories. To ensure the robustness of our evaluation, we employ four well-established pretrained classification models as benchmarks. The selected pretrained models include variants of ResNet (He et al., 2016) (ResNet18 and ResNet50) and Vision Transformer (ViT) (Dosovitskiy, 2020) (ViT-B/16 and ViT-B/32). These models represent state-of-the-art approaches in image recognition.

#### 4.1.1 SYNTHETIC DATASET FOR TOKEN-BASED ANALYSIS

To elucidate the selected $\mathcal{T}^*$ from our proposed explanation method $\phi$ of VQGMs, we use a synthetic data set generated by VQGAN (Esser et al., 2021), which encompasses the same categories as ImageNet. This symthetic dataset allows us to directly examine how the generative model utilizes tokens to encode concept-specific information, specifically class labels, rather than analyzing real-world images. Our synthetic dataset consists of 1,000,000 training images, 300,000 validation images, and 50,000 test images, evenly distributed across all ImageNet categories. This balanced distribution provides 1,000, 300, and 50 images per category in the training, validation, and test sets, respectively. Each generated image has $256 \times 256$ pixels, with $16 \times 16$ tokens. During the generation process, we can simultaneously obtain both the image and its token-based embedding $\mathbf{E}$.

#### 4.1.2 TRAINING INFORMATION EXTRACTOR

To validate the reliability of our explanations, we train 2 classification models with identical input-output specifications but different architectures as the information extractor $f_Y$: (1) CNN-based Extractor (CE), and (2) ResNet-based Extractor (RE). The specific architectures can be found in the Appendix A.1. Both models take token-based embeddings $\mathbf{E} \in \mathbb{R}^{256 \times 16 \times 16}$ and output probability distributions over 1000 ImageNet labels. This setup enables us to analyze how different information extractor architectures generate $T^*$ based on the VQGAN, ensuring that our explanations are consistent across model designs.

### 4.2 INFORMATION EXTRACTOR PERFORMANCE EVALUATION

Based on Tabel 1, the performance of our models, with Top 1 accuracies of 53.07% and 51.43% for CNN-based and ResNet-based architectures, respectively, is significant considering the potential inaccuracies introduced by VQGAN image generation process. Despite this challenge, the high Top 5 accuracies (exceeding 75%) demonstrate that our information extractors effectively capture the relationship between token patterns and image labels. This indicates the models' ability to learn meaningful representations from token-based input embeddings $\mathbf{E}$.

### 4.3 SALIENCY-BASED TOKEN EXPLANATION EVALUATION

**Experimental Design.** In this part, our experiments aim to validate that the token combinations $\mathcal{T}^*_{\text{image}}$ and $\mathcal{T}^*_{\text{concept}}$ identified by our saliency-based method are indeed the most relevant and representative of specific concepts. We conduct evaluations at both the image and the concept level.

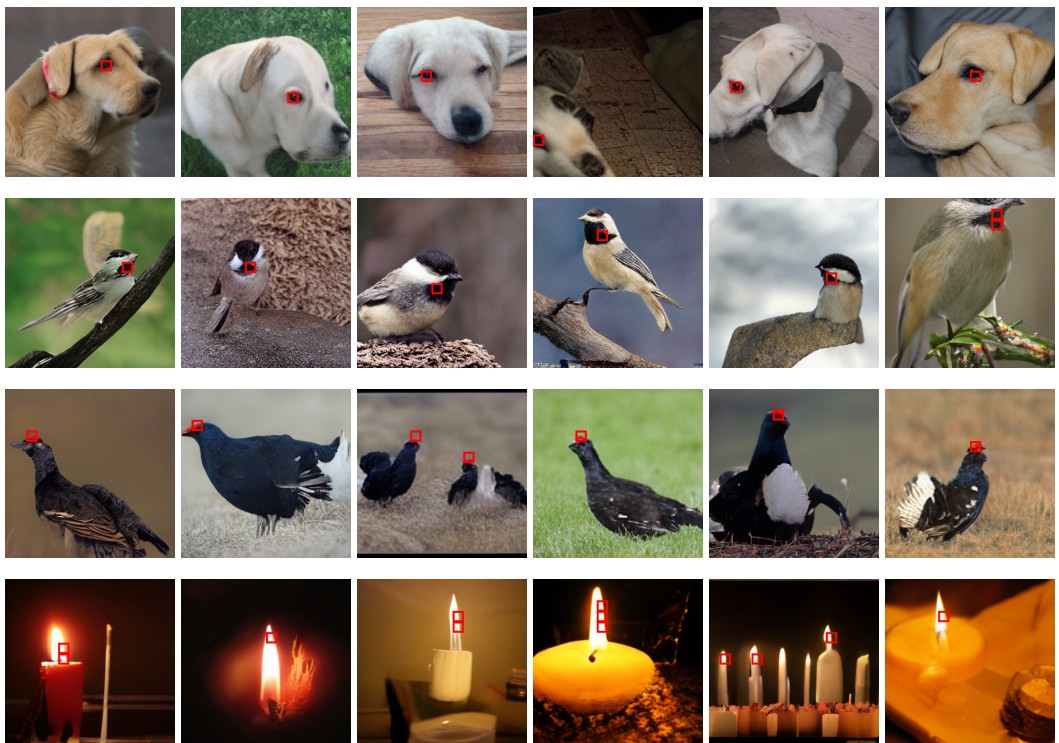

Figure 2: Token visualization for different concepts using our saliency-based method. Each row shows 6 images of a distinct concept. Red boxes highlight high-TSV tokens, revealing consistent identification of class-specific features (e.g., eyes, neck, red crowns, flame) across images.

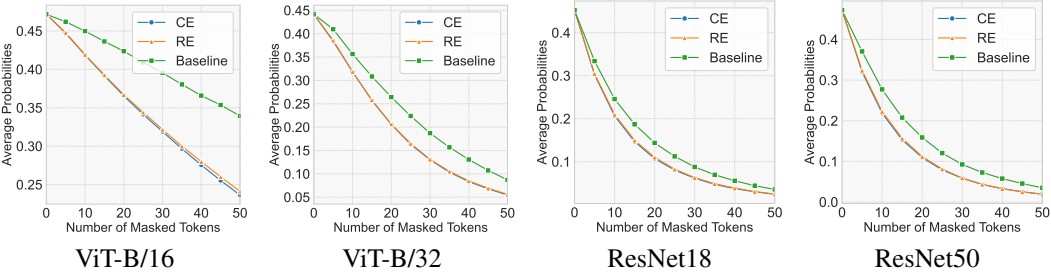

Figure 3: Image-level saliency-based method evaluation results: average probabilities vs. number of masked tokens.

At the image level, we define $\mathcal{T}_{\text{image}}^*$ as the set of the Top $n$ tokens with the highest Token Saliency Values (TSV), where $n$ ranges from 5 to 50 in increments of 5. For each token in $\mathcal{T}_{\text{image}}^*$, we mask the corresponding regions in the generated image. We then measure the change in softmax probability for the specific label across four pretrained models: ViT-B/16, ViT-B/32, ResNet18, and ResNet50. As a baseline, we randomly select $n$ tokens and mask the associated regions in the image to compare the effect with masking tokens in $\mathcal{T}_{\text{image}}^*$. At the concept level, we first identify the Top n ($n = 20$) highest-TSV tokens in each training image to form individual $\mathcal{T}_{\text{image}}^*$ sets. From the union of these sets across all images of a given concept, we select the Top k ($k = 100$) most frequent tokens to form $\mathcal{T}_{\text{concept}}^*$. We compare this with a frequency-based baseline of selecting the Top 100 most frequent tokens in all images under the specific label without using our information extractor $f_Y$. We then mask the corresponding patches of selected tokens in test images and measure the change in the probability of the label. Our analyses treat each ImageNet class as a distinct concept, aggregating results across all 1000 classes.

Table 2: Concept-level saliency-based method evaluation results (Acc: prediction accuracy, P: probability, $n$: number of masked tokens, $\Delta A$: change in accuracy, $\Delta P$: change in probability).

| Pretrained Model | Acc | P | CE | | | RE | | | Baseline | | |
|---|---|---|---|---|---|---|---|---|---|---|---|
| | | | $n$ | $\Delta A\downarrow$ | $\Delta P\downarrow$ | $n$ | $\Delta A\downarrow$ | $\Delta P\downarrow$ | $n$ | $\Delta A\downarrow$ | $\Delta P\downarrow$ |
| **ResNet18** | 55.6% | 0.452 | | -45.4% | -0.381 | | **-45.5%** | **-0.382** | | -42.2% | -0.356 |
| **ResNet50** | 56.1% | 0.472 | 42.176 | -46.6% | -0.403 | 40.629 | **-46.8%** | **-0.404** | 64.166 | -41.9% | -0.365 |
| **ViT-B/16** | 59.0% | 0.472 | | -9.50% | -0.112 | | **-9.60%** | **-0.113** | | -7.30% | -0.090 |
| **ViT-B/32** | 58.0% | 0.442 | | **-36.0%** | **-0.289** | | **-36.0%** | **-0.289** | | -33.2% | -0.264 |

**Image-level Evaluation Results.** Figure 2 demonstrates the efficacy of our saliency-based method in identifying concept-specific features across multiple images (more results can be found in Appendix A.3). Each row in the figure represents a distinct label. For each label, we present 6 different images. Within each image, we highlight a specific token that exhibits a high Token Saliency Value (TSV) using a red bounding box. These visualizations demonstrate our saliency-based method's ability to focus on tokens that often correspond to specific, concrete visual features within each concept. For instance, the consistent highlighting of eyes, red crowns, and other distinctive features across multiple images of the same class indicate that these tokens can effectively represent meaningful, class-specific characteristics.

Quantitatively, Figure 3 shows the average change in softmax probability for specific labels as we mask from 5 to 50 high-TSV tokens. Across all pretrained models, our method consistently leads to a steeper decline in probability compared to random selection, demonstrating its effectiveness in identifying tokens crucial to concept representation. Notably, both CNN-based and ResNet-based information extractors exhibit similar declining trends, suggesting that different models attend to similar tokens for specific concepts. These results validate our saliency-based method's ability to identify label-relevant features and provide interpretable insights into VQGMs.

**Concept-level Evaluation Results.** Table 2 presents the concept-level evaluation results across 4 different pretrained models. Our saliency-based method consistently outperforms the naive frequency-based baseline in terms of impact on model predictions. When removing tokens based on our $\mathcal{T}^*_{\text{concept}}$, we observe a more substantial impact on the prediction of the pre-trained model despite masking fewer tokens on average ($n_1 = 42.176$ and $n_2 = 40.629$ for the CNN-based extractor and the ResNet-based extractor, respectively) compared to the baseline ($n_b = 64.166$). This is evidenced by the more significant reductions in both accuracy and probability scores. For instance, with ResNet50, our method decreases accuracy by $46.8\%$ and probability by $0.404$, compared to baseline $41.9\%$ and $0.365$, respectively. This superior performance, despite masking a smaller image area, indicates that our method identifies more label-relevant tokens. Despite masking more tokens, the baseline's inferior performance suggests that it often selects less relevant tokens. These results validate the effectiveness of our information extractor in identifying concept-specific features and demonstrate our method's ability to capture more semantically meaningful tokens for each concept.

## 4.4 OPTIMIZATION-BASED TOKEN EXPLANATION EVALUATION

**Experimental Design.** To evaluate our optimization-based method for identifying concept-specific tokens $\mathcal{T}^*_{\text{concept}}$, we conduct experiments using 10 bird categories (500 images in total) from the synthetic dataset. These bird images can be generated by VQGAN in high quality, and the 4 pre-trained models achieve high prediction accuracies on these 500 images, ranging from $84.6\%$ to $90.2\%$ (refer to $\text{Acc}_{\text{Orig}}$ in Table 3). We pair 10 bird categories into 5 groups, each category serving as both original and target labels. The optimization process begins with token-based embeddings $\mathbf{E}$ from original label images, optimizing towards the target label, which is the concept to be explained (details in Appendix A.4). We focus on a fixed $4 \times 4$ region within the $16 \times 16$ token grid, limiting $\mathcal{T}^*_{\text{concept}}$ to 16 tokens (only $1/16$ of total 256 tokens).

We explore two optimization methods, both aiming to maximize the activation of a target bird label (the concept to be explained): 1) *Token selection optimization (our method)*: We optimize a token selection matrix, which represents the probability of selecting each token from the codebook for specific positions in the target region. 2) *Embedding optimization (baseline)*: We directly optimize the $d$-dimensional embedding in the target region. After optimization, we apply vector quantization Gray (1984) to map each optimized embedding vector to the nearest token in the codebook,

Table 3: Optimization-based method evaluation results.

| Model | Acc$_{\text{Orig}}$ | P$_{\text{Orig}}$ | P$_{\text{Targ}}$ | $\Delta$P$_{\text{Orig}}\downarrow$ / $\Delta$P$_{\text{Targ}}\uparrow$ | | | |
| | | | | Embedding Optimization | | Token Selection | |
| | | | | CE | RE | CE | RE |
|---|---|---|---|---|---|---|---|
| **ResNet18** | 86.8% | 0.795 | 0.016 | -0.224 / 0.127 | -0.202 / 0.097 | **-0.290 / 0.162** | -0.284 / 0.160 |
| **ResNet50** | 84.6% | 0.780 | 0.011 | -0.217 / -0.122 | -0.206 / 0.102 | **-0.282 / 0.165** | -0.277 / 0.155 |
| **ViT-B/16** | 89.0% | 0.766 | 0.003 | -0.193 / 0.095 | -0.174 / 0.072 | -0.237 / **0.128** | **-0.243** / 0.121 |
| **ViT-B/32** | 90.2% | 0.758 | 0.003 | -0.196 / 0.090 | -0.190 / 0.080 | -0.239 / 0.112 | **-0.245 / 0.120** |

minimizing the Euclidean distance. Optimized embeddings were decoded via VQGAN to generate images. To measure the effectiveness of each method optimization method, we calculate the change in probabilities for the original and target labels before and after optimization:

$$\Delta P_{\text{Orig}} = P_{\text{Orig}}(\text{optimized}) - P_{\text{Orig}}(\text{initial}), \quad \Delta P_{\text{Targ}} = P_{\text{Targ}}(\text{optimized}) - P_{\text{Targ}}(\text{initial}). \quad (10)$$

**Evaluation Results.** Table 3 shows the results of our optimization methods across different models. The *Token Selection* method consistently outperforms the *Embedding Optimization* baseline by both reducing the original label probability ($\Delta P_{\text{Orig}}$) and increasing the target label probability ($\Delta P_{\text{Targ}}$). For instance, in the ResNet18 model with CNN-based extractor, our method decreases the probability of the original label by $36.5\%$ (from $0.795$ to $0.505$) and increases the probability of the target label about 11 times (from $0.016$ to $0.178$) compared to the initial probability. In contrast, the Embedding Optimization baseline achieves a $28.2\%$ decrease in the original label probability and a 7.9-fold increase in the target label probability. This shows that our method surpasses the baseline by achieving a greater reduction in the original label and a more significant increase in the probability of the target label. Similar improvements are observed in another information extractor. In the ViT-B/16 model with ResNet-based extractor, our Token Selection method reduces the probability of the original label by $31.7\%$ and increases the probability of the target label by over 40 times, significantly outperforming the baseline. These results indicate that our Token Selection method effectively identifies the most important tokens contributing to the target label. By directly optimizing the token selection matrix end-to-end, it finds the token combination that maximally activates our information extractor $f_Y$. In contrast, the Embedding Optimization method optimizes embeddings and then maps them back to the nearest tokens in the codebook, which may result in suboptimal token combinations due to the lack of end-to-end optimization.

These quantitative results demonstrate our optimization method's capability to capture token-level differences between concepts/labels and suggest its potential for targeted image manipulation. To further illustrate this capability visually, we conduct image editing experiments. Figure 4 shows the visualization of the image editing process using our token selection optimization method. These sequences show the gradual transformation of one bird species into another, focusing on the head region. The progressive changes in the bird's head features, such as beak shape and color, illustrate our method's ability to identify and manipulate concept-specific tokens effectively.

These results validate the effectiveness of our token selection method in identifying and manipulating class-relevant features within VQGMs. The substantial increases in target label probabilities, often by more than an order of magnitude, demonstrate the method's potential for enhancing model interpretability and its applicability in targeted image editing tasks.

## 4.5 DISCUSSION

Our experiments demonstrate that CORTEX effectively interprets VQGMs by identifying and manipulating concept-specific tokens within the codebook, thereby revealing how these tokens contribute to the model's representation of concepts. The saliency-based method accurately locates tokens critical for specific concepts, while the optimization-based method enables targeted manipulation of these tokens to achieve controlled, concept-driven modifications in generated images. This elucidation of the relationship between codebook tokens and encoded concepts enhances our understanding of VQGMs' internal representations and decision-making processes, which is crucial for identifying potential biases and improving model robustness.

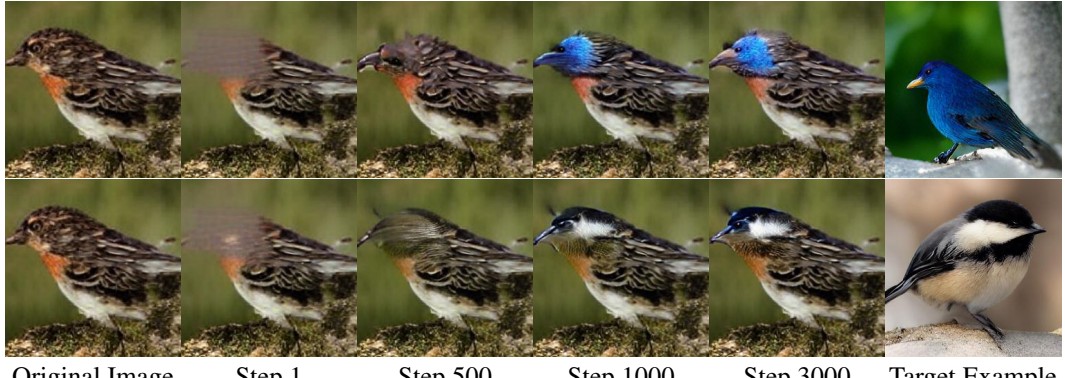

Original Image     Step 1     Step 500     Step 1000     Step 3000     Target Example

Figure 4: Optimization-based image editing process.

## 5 RELATED WORK

**Vector quantization in computer vision.** Vector quantization has been instrumental in advancing image generative models (Gray, 1984; Nasrabadi & Feng, 1988). VQ-VAE (Van Den Oord et al., 2017) pioneered the use of discrete latent codes for efficient image reconstruction, overcoming "posterior collapse" issues in VAEs. DALL-E (Ramesh et al., 2021) extended this to text-to-image generation, while VQGAN (Esser et al., 2021) and ViT-VQGAN (Yu et al., 2021) enhanced image quality through perceptual and adversarial objectives. In video generation, MAGVIT (Yu et al., 2023), VideoPoet (Kondratyuk et al., 2023), and LaVIT (Jin et al., 2024; 2023) applied vector quantization for spatial-temporal modeling and multimodal learning. Our work builds upon these VQGMs, offering a novel approach to interpreting discrete tokens and providing insights into visual information encoding and utilization.

**Vision model explainability.** Traditional approaches to explaining vision models primarily fall into two categories: heatmap-based methods (Sundararajan et al., 2017; Selvaraju et al., 2020; Binder et al., 2016; Gandelsman et al., 2023; Chefer et al., 2021), which highlight influential image regions, and optimization-based methods (Nguyen et al., 2016; Erhan et al., 2009; Yosinski et al., 2015; Nguyen et al., 2015; Simonyan, 2013), which generate synthetic inputs to maximize specific activations. While insightful, these pixel-level approaches are limited in explaining complex generative models like VQGMs. Our CORTEX approach extends these ideas to the token level, providing concept-specific explanations of how VQGMs utilize discrete latent representations. This novel perspective offers deeper insights into the internal generative processes of VQGMs, bridging the gap between traditional explainability methods and the complexities of modern generative models.

## 6 CONCLUSION

In this paper, we introduce CORTEX, a novel framework for interpreting VQGMs through concept-specific token analysis. Guided by the Information Bottleneck principle, CORTEX combines saliency-based and optimization-based methods to extract critical token combinations while filtering out non-essential tokens. Our comprehensive experiments demonstrated CORTEX's effectiveness in providing clear explanations of token combination in the generative process, significantly outperforming baselines. The saliency-based method accurately identified tokens critical for specific concepts, while the optimization-based method enabled targeted manipulation of these tokens to achieve controlled, concept-driven modifications in generated images.

This elucidation of the relationship between codebook tokens and encoded concepts enhances our understanding of VQGMs' internal representations and decision-making processes. CORTEX not only improves VQGM transparency but also enables precise image manipulation and editing based on token-level representations. This work opens up new possibilities for fine-grained control in image generation and editing, paving the way for more transparent and controllable generative models. Future work could explore extending CORTEX to other generative architectures and investigating its potential in multi-modal settings.

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

# A APPENDIX

## A.1 INFORMATION EXTRACTOR

This appendix provides details on the structure of two Information Extractor: CNN-based model and Resnet-based model.

### A.1.1 CE: CNN-BASED EXTRACTOR

The CNN-based Extractor (CE) is a convolutional neural network designed for image classification. The model comprises two main blocks, each containing four convolutional layers (conv1_1 to conv1_4 and conv2_1 to conv2_4). Each convolutional layer utilizes $512$ filters with a $3 \times 3$ kernel size, stride of 1, and padding of 1, followed by batch normalization and ReLU activation. Max pooling ($2 \times 2$ kernel, stride 2) is applied after each block. The network concludes with three fully connected layers: the first transforms $512 \times 4 \times 4$ input features to $4096$ output features, the second maintains $4096$ features, and the final layer maps to the number of classes. Additional features include batch normalization and ReLU activation after the first two fully connected layers, with dropout $(0.5)$ applied after the first fully connected layer.

### A.1.2 RE: RESNET-BASED EXTRACTOR

The ResNet-based Extractor (RE) is an advanced model incorporating residual connections and Squeeze-and-Excitation (SE) blocks. The network consists of two main layers, each containing 3 residual blocks. Each residual block comprises two convolutional layers ($3 \times 3$ kernel, maintaining channel size) with batch normalization and ReLU activation, a shortcut connection, and an SE block for channel-wise attention. The SE block employs global average pooling followed by two fully connected layers with reduction and expansion, using sigmoid activation for generating attention weights. The model concludes with global average pooling to reduce spatial dimensions, followed by two fully connected layers: $512$ to $2048$ features, and $2048$ to the number of classes. Batch normalization and ReLU activation are applied after the first fully connected layer, with dropout $(0.5)$ implemented.

Both CE and RE are designed to process input tensors with 256 channels and spatial dimensions of $16 \times 16$ tokens.

### A.1.3 TRAINING SETTINGS

These information extractors were trained using a batch size of 256 for 80 epochs, with the task involving classification across 1000 classes. We employed the Adam optimizer with an initial learning rate of $0.001$ and weight decay of $1e-4$. To adjust the learning rate during training, we implemented a StepLR scheduler, which decreased the learning rate by a factor of $0.1$ every 20 epochs. The loss function used for training was Cross Entropy Loss. Our experimental setup allowed for the training of multiple model architectures under consistent conditions, enabling fair comparison of their performance.

## A.2 GUMBEL-SOFTMAX TECHNIQUE FOR TOKEN SELECTION OPTIMIZATION

In our implementation, we employ the Gumbel-Softmax technique to optimize the selection of tokens from the codebook. This method enables differentiable sampling from a discrete distribution, which is essential for our gradient-based optimization process. The core of our approach involves a matrix P of shape $(256, 16384)$, where each row represents a probability distribution over the codebook tokens.

The Gumbel-Softmax approximation operates by adding Gumbel noise to the logits (log probabilities) derived from P at each optimization step. The Gumbel-Max trick states that for a categorical distribution with class probabilities $p_i$, sampling can be performed as:

$$\text{argmax}_i(\log(p_i) + g_i) \tag{11}$$

where $g_i$ are i.i.d. samples from Gumbel(0, 1) distribution.

We then apply a softmax function with a temperature parameter $\tau$ to these noisy logits:

$$y_i = \frac{\exp((\log(p_i) + g_i)/\tau)}{\sum_j \exp((\log(p_j) + g_j)/\tau)} \tag{12}$$

In the "hard" version of this technique, we convert this soft distribution to a one-hot vector by selecting the maximum value:

$$y_{\text{hard}} = \text{onehot}(\text{argmax}_i(y_i)) \tag{13}$$

The final output is then:

$$y = \text{stop\_gradient}(y_{\text{hard}} - y) + y \tag{14}$$

This process allows us to sample discrete tokens while maintaining differentiability, thereby enabling backpropagation through the sampling process.

A key feature of the Gumbel-Softmax is the temperature parameter $\tau$, which controls the discreteness of the samples. As $\tau$ approaches zero, the samples become more discrete, closely approximating one-hot vectors. Conversely, as $\tau$ increases, the distribution becomes smoother and more uniform.

Throughout the optimization process, we update the P matrix based on the gradients computed through this differentiable sampling procedure. The gradient of the Gumbel-Softmax estimator with respect to the logits is:

$$\frac{\partial y_i}{\partial \log(p_k)} = \frac{y_i(\delta_{ik} - y_k)}{\tau} \tag{15}$$

where $\delta_{ik}$ is the Kronecker delta.

By utilizing this approach, we can optimize the selection of discrete tokens from the codebook in a manner compatible with gradient-based optimization methods. This compatibility is crucial for our objective of maximizing the activation of target labels in our classification model.

The Gumbel-Softmax technique thus serves as a bridge between the discrete nature of our token selection problem and the continuous optimization landscape required for effective gradient-based learning. It allows us to backpropagate through the discrete sampling operation, enabling end-to-end training of our model while maintaining the ability to produce discrete outputs during inference.

### A.3 MORE VISUALIZATOIN RESULTS

Figure 5 presents additional image-level visualization results, complementing the analysis provided in Section 4.3. The figure is structured into two distinct sets of four rows each, each set focusing on a specific token for a particular category. This approach demonstrates the efficacy of our saliency-based method in identifying concept-specific features across multiple images. The first four rows showcase visualizations related to the "black grouse" category, highlighting a single, consistently meaningful token across different images of this bird species. Similarly, the subsequent four rows are dedicated to visualizations of the "candle" category, emphasizing the same token across various candle images. In each image, we highlight the token exhibiting a high Token Saliency Value (TSV) using a red bounding box. These visualizations illustrate our saliency-based method's ability to focus on tokens that frequently correspond to specific, concrete visual features within each concept. For instance, the consistent highlighting of particular features (such as the red crown for the black grouse or flame for candles) across multiple images of the same class indicates that these tokens effectively represent meaningful, class-specific characteristics. By consistently focusing on the same token within each category, we demonstrate our method's ability to extract and emphasize stable, category-relevant features across diverse visual representations.

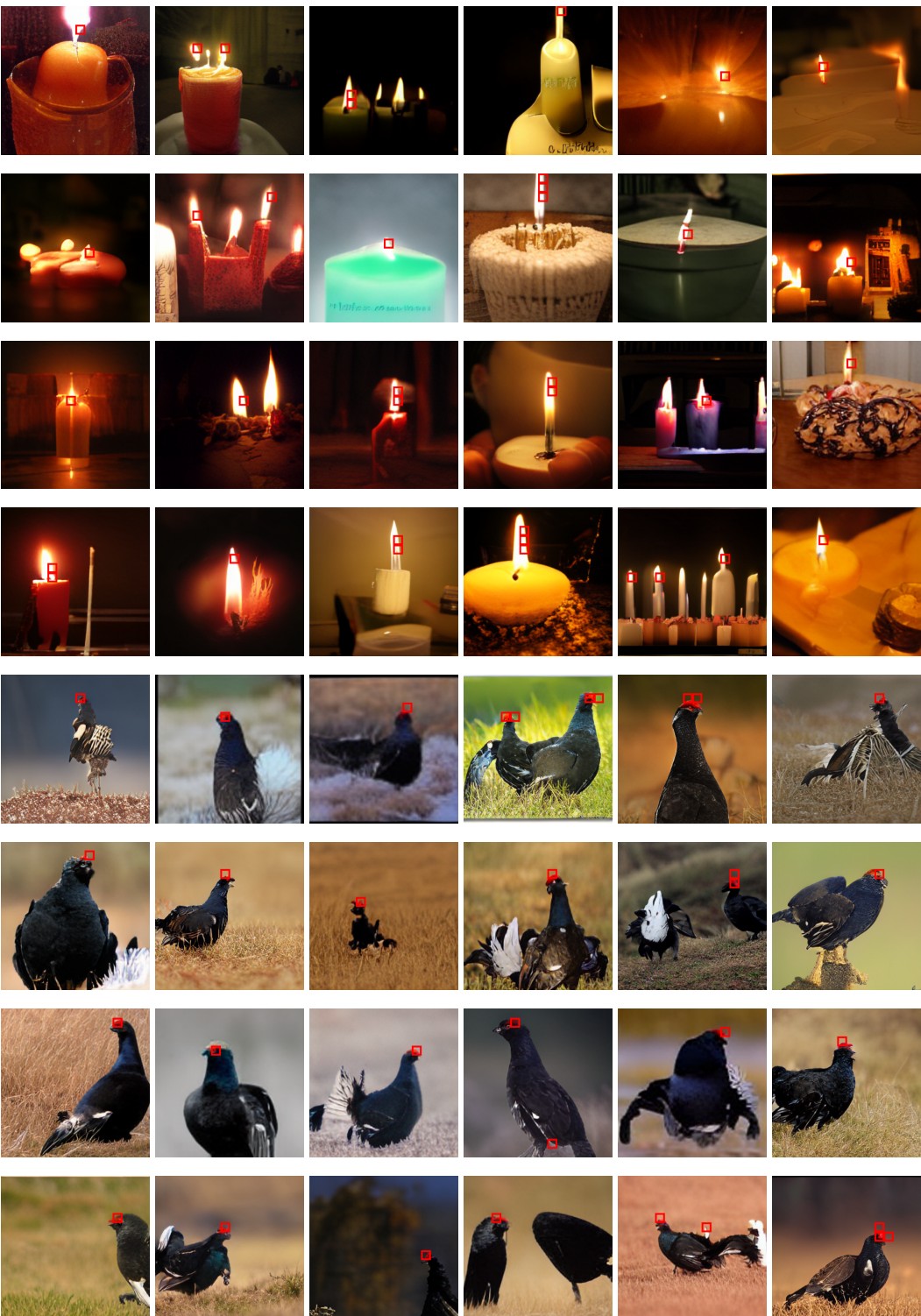

Figure 5: Token visualization in different categories: red box in each row represents the same token

## A.4 OPTIMIZATION EXPERIMENT DESIGN

Figure 6 illustrates the pairing strategy employed in our optimization experiments. We selected 10 bird categories and organized them into 5 pairs, as shown in the figure. Each pair consists of two bird species that serve alternately as the original and target labels in our experiments.

For each category, we utilized 50 images from the test set. In the experimental process, when images from a category in the first row serve as the original images, the corresponding category in the second row becomes the target label, and vice versa. For instance, in the pair (Goldfinch, Water Ouzel), when Goldfinch images are being optimized, Water Ouzel serves as the target label. Conversely, when Water Ouzel images are used as original images, Goldfinch becomes the target label.

This reciprocal design is applied consistently across all five pairs depicted in Figure 6. Every image in our dataset undergoes optimization as an original image, with its paired category serving as the target label.

Our optimization process focused on a fixed $4 \times 4$ region within the $16 \times 16$ token grid, limiting $\mathcal{T}^*_{\text{concept}}$ to a set of 16 tokens. We evaluate the changes in softmax probabilities for both the original and target labels across four pretrained models. This figure shows that

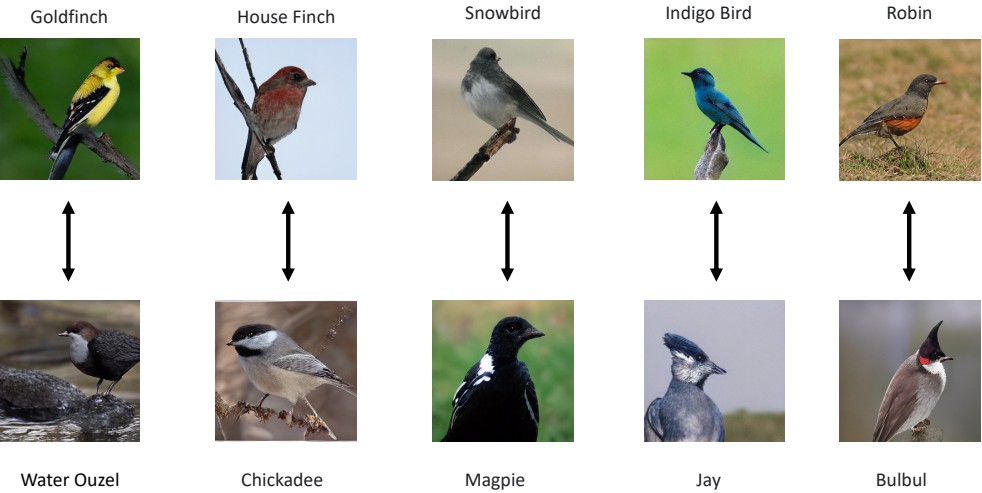

Figure 6: Optimization-based method experiment design

