# OpenReview forum: "CORTEX: Concept-Oriented Token Explanation in Vector-Quantized Generative Model"
_ICLR.cc/2025/Conference — Submitted to ICLR 2025_

### Official Review · Reviewer_oWeq · 2024-10-30

**Soundness:** 2
**Presentation:** 2
**Contribution:** 2
**Rating:** 3
**Confidence:** 3

**Summary:**

The work looks to expand explainability and interpretability methods to vector-quantized generative models. The authors spend little time motivating the work, but the goal seems to be to discover how the discrete tokens used by VQGMs can be meaningfully interpreted in deployed models on a concept level. The authors propose a token based saliency-based information bottleneck in conjunction with a proposed input independent method for codebook search in an input dependent and input independent approach, respectively.

Briefly summarised; the input dependent framework applies a gradient based saliency method (Simonyan et al., 2013) to retrieve a token saliency value (TSV). Then, by using a support set of conceptually related images to mine for highly frequent tokens related to a specific concept. For the input-independent approach, tokens are mined using a token selection matrix with gradient based discrete optimization with the GumbelSoftmax.

Overall, the work unfortunately comes across as somewhat niched. In conjunction with the limited motivation by the authors, this reviewer has difficulty placing the work in the broader context of current research, neither in the field of explainability nor generative image modelling. Showing how the approach could be applied in more general context could potentially strengthen the work.

**Strengths:**

S1: The methodology of combining saliency and optimization methods for selecting relevant tokens in codebook approaches has a degree of novelty, which could possibly generalise to other applications, such as mining concept based prototypes. This would possibly also allow for more baselines, as this field is an active field of study with several contributions in active use and development.

S2: Targeted editing with VQGMs have applications in content moderation and interactive generation. The proposed method’s basis in delineating regions by concept seems appropriate in these tasks.

**Weaknesses:**

W1: The saliency method is based on Simonyan et al. 2013, which has been shown to be highly independent of model predictions. Adebayo et al. 2018 show that saliency maps produced by these methods demonstrate little change when replacing most of the layers of a network under scrutiny with randomly initialised weights, hence the method the current work bases its saliency method on has dubious interpretive value.

W2: The use of a tailored synthetic dataset generated as an evaluation of the proposed method strikes this reviewer as dubious, even given the setting of interpretations for generative models. This limits the overall generalisability of the approach to be specific for VQGMs, as opposed to a more general mining approach. If this was a necessity for training the method (which is not clear to this reviewer at this point), the argument for its use in evaluation should be emphasised in the manuscript.

W3: The value of the proposed method comes across as highly specific and somewhat niched. This necessarily translates to limited context and baselines in other works.

W4: Aside from targeted image editing with VQGMs, the fundamental motivation behind the work is insufficiently established in the introduction. The importance of interpretability in VQGMs could be more strongly emphasised in the manuscript. Consider demonstrating the motivation behind the work to the reader through examples or potential applications.

W5: The overall methodology is, in this reviewer's opinion, somewhat unclear. This could be improved by highlighting the importance of input dependence / independence, discussing how these two approaches interact, and more explicitly clarifying or showing how they are combined to form a unified interpretation with concepts.

W6: The proposed method is designed specifically for VQGMs, and as such, has somewhat limited scope. As diffusion models have been shown to generally provide higher quality in image generation, the method is naturally more limited in impact. While this in and of itself does not mean the work does not have value, considering the impact plays a role in evaluating the work for the conference.

**Questions:**

Q1: What are the authors opinions on the broader impact of the work, and what concrete problems their proposed methods solve, aside from targeted editing with VQGMs?

Q2: Could the reliance on the synthetically generated dataset be dropped? If not, could the authors provide a convincing rationale for its necessity?

Q3: Can mined concepts from the codebook be applied in other applications with discrete token-based backbones?

Q4: Given aforementioned issues with Simonyan et al., did the authors consider ablating the effects of token based saliency with alternative approaches?

---

> ### Comment · Reviewer_oWeq · 2024-11-26
> **No rebuttal from authors?**
>
> I would like to remind the authors that the rebuttal phase is coming to a close, and as of now, reviewers have yet to recieve any replies to the published reviews.
>
> If the authors look to partake in any meaningful discussion on their work, responding to the reviews are necessary. For my own sake, I would appreciate some time digesting their responses to be able to make a final decision on their work.

---

> ### Author Response · Authors · 2024-11-27
> **Thank you! (1/2)**
>
> [**W1**, **Q4**]: *The saliency method based on Simonyan et al. (2013) has dubious interpretive value due to issues highlighted by Adebayo et al. (2018).*
>
> **Response:**  In our paper, visualization is not used as an evaluation metric. Instead, we focus on quantitative comparisons, as shown in the experiments in Table 2, where we demonstrate the effectiveness of our approach by masking tokens with high Token Saliency Values (TSV) and comparing them to masking tokens based on frequency.
>
> This experiment clearly shows that our information extractor can identify tokens strongly correlated with the target concept.
>
> Moreover, it is important to note that the information extractor is not the model we aim to interpret. Our experiments are not designed to prove that the information extractor perfectly encapsulates all token information. The information extractor is adaptable and can be tailored to different concepts based on user selection, serving as a flexible tool for explanation.
>
> The role of TSV in our work is to validate the effectiveness of our approach. Specifically, it shows that a small external model (the information extractor) can interpret high-dimensional, complex token information that is otherwise difficult for humans to understand. Our paper's key contribution lies in demonstrating the effectiveness of this method rather than proving the comprehensiveness of the information extractor itself.
>
>
> [**W2**, **Q2**]: *The use of a synthetic dataset may limit generalizability. Could the reliance on this dataset be dropped?*
>
> **Response:** Our method does not rely exclusively on synthetic datasets and can be applied to real-world datasets.
>
> The primary reason we used a synthetic dataset generated by VQGM is to analyze the behavior of the VQGM. By focusing on model-generated data, we can better understand how the VQGM encodes and expresses these concepts. This emphasis aligns with our goal of explaining the VQGM’s behavior, which is why we ultimately chose to analyze data generated by the model.
>
> Additionally, in our preliminary experiments, we tested the approach using real datasets. By encoding existing images through a pretrained encoder and using the vector quantization step to identify the closest codebook token for each embedding, we can generate the inputs required for the information extractor described in our paper. However, we ultimately chose to use generated data because using real datasets explains the distribution of the concept in real images rather than the model's understanding of the concept, which is the focus of our study.
>
> **[W3]**: *The value of the proposed method seems niche and specific, with limited context and baselines.*
>
> **Response:** We would like to clarify that the proposed method is broadly applicable and not limited to a specific niche. Currently, generative models primarily fall into two categories: diffusion-based models and vector-quantized generative models (VQGMs). Our approach is designed to work with any VQGM that employs a discrete codebook, regardless of the number of concepts or the size of the codebook.
>
> This interpretability framework is equally applicable to text-to-image VQGMs, where the concepts can be defined as the context provided in the model's prompt. By training a lightweight information extractor tailored to the specific concept of interest, token-based concept explanations can be generated.
>
> To the best of our knowledge, our method is the first to interpret tokens in the codebook. As such, our method is versatile, capable of interpreting any model that utilizes a codebook, and provides a generalizable approach for explaining the learned representations of any concept. This broad applicability makes the method relevant to a wide range of applications beyond the specific use case presented in our work.
>
> [**W4**, **Q1**]: *The fundamental motivation behind the work is insufficiently established. The importance of interpretability in VQGMs could be more strongly emphasized.*
>
> **Response:** We appreciate this feedback. Our work enhances the interpretability and transparency of generative models, aiding in debugging, bias detection, and ethical use. By revealing how concepts are encoded at the token level, developers can improve model design, control the generative process, and ensure intended behavior. Our methods address challenges such as mitigating biases, improving robustness to adversarial attacks, and enabling content customization. By providing insights into concept representation, our approach promotes responsible use, enhances transparency, and fosters user trust.

---

> ### Author Response · Authors · 2024-11-27
> **Thank you! (2/2)**
>
> **[W5]** *The methodology is somewhat unclear. Highlighting the importance of input dependence/independence and how the two approaches interact could improve clarity.*
>
> **Response:** The saliency-based method provides fine-grained, instance-specific explanations by identifying tokens crucial for a concept in a particular image. In contrast, the optimization-based method offers a global perspective by finding token combinations representing a concept across all possible images, independent of specific inputs.
>
> Under circumstances where an explanation is needed for a specific generated image, the saliency-based method is more informative. When seeking a general understanding of how a concept is represented in the model's codebook, the optimization-based method is more appropriate.
>
> Therefore, the saliency-based method interprets tokens at the level of specific generated images, while the optimization-based method explains the entire codebook. Additionally, the results of the optimization-based method can also be used for tasks such as image editing.
>
> **[W6]** *The method is specific to VQGMs, and diffusion models have shown higher quality in image generation, potentially limiting the impact of the work.*
>
> **Response:** While diffusion models are indeed powerful, VQGMs offer advantages such as efficient discrete representations and are still widely used in practice [1, 2, 3, 4, 5, 6, 7]. Furthermore, our methods can potentially be adapted to interpret discrete components within diffusion models or other generative models that incorporate quantization.
>
> We will discuss this possibility in the revised paper, highlighting how our approach can contribute to the broader field of generative modeling, including models that combine discrete and continuous representations.
>
> **[Q3]** *Can mined concepts from the codebook be applied in other applications with discrete token-based backbones?*
>
> **Answer:** Yes, our approach is broadly applicable to any model that uses discrete tokens as part of its architecture. The core idea is to leverage a small external model (information extractor) to interpret high-dimensional token representations that are otherwise difficult for humans to understand. This makes the framework highly adaptable to other domains.
>
> For example, in natural language processing (NLP), models with discrete vocabularies can benefit from similar analysis. By training an information extractor tailored to specific concepts in the NLP domain, token-level contributions to particular outputs can be effectively analyzed and interpreted. This demonstrates the versatility of our framework and its potential applications beyond the context of generative models.
>
> [1] Esser, Patrick, Robin Rombach, and Bjorn Ommer. "Taming transformers for high-resolution image synthesis." *Proceedings of the IEEE/CVF conference on computer vision and pattern recognition*. 2021.
>
> [2] Wang, Xinlong, et al. "Emu3: Next-token prediction is all you need." *arXiv preprint arXiv:2409.18869* (2024).
>
> [3] Yu, Lijun, et al. "Magvit: Masked generative video transformer." *Proceedings of the IEEE/CVF Conference on Computer Vision and Pattern Recognition*. 2023.
>
> [4] Yu, Lijun, et al. "Language Model Beats Diffusion--Tokenizer is Key to Visual Generation." *arXiv preprint arXiv:2310.05737* (2023).
>
> [5] Kondratyuk, Dan, et al. "Videopoet: A large language model for zero-shot video generation." *arXiv preprint arXiv:2312.14125* (2023).
>
> [6] Jin, Yang, et al. "Unified language-vision pretraining with dynamic discrete visual tokenization." *arXiv preprint arXiv:2309.04669* (2023).
>
> [7] Jin, Yang, et al. "Video-lavit: Unified video-language pre-training with decoupled visual-motional tokenization." *arXiv preprint arXiv:2402.03161* (2024).

---

> > ### Comment · Reviewer_oWeq · 2024-11-29
> >
> > ## Response to Comments
> >
> > ### [W1, Q4]
> > > [W1, Q4] In our paper, visualization is not used as an evaluation metric. Instead, we focus on quantitative comparisons...
> >
> > It is important to note that the results in Adebayo et al. (2018) is not limited to qualitative visualization, and extends to quantitative results. The issue is that gradient based saliency generally produces results independent of whether the network weights are trained or not. This reviewer is not convinced that your outlined approach circumvents this issue.
> >
> > ###  [W2, Q2]
> > > [W2, Q2] Our method does not rely exclusively on synthetic datasets and can be applied to real-world datasets...  in our preliminary experiments, we tested the approach using real datasets.
> >
> > Including details of these experiments in the manuscript or the appendix would serve to alleviate concerns on generalizability.
> >
> > ### [W3, W4, Q1, Q3]
> > > [W3] We would like to clarify that the proposed method is broadly applicable and not limited to a specific niche.
> >
> > > [Q3] our approach is broadly applicable to any model that uses discrete tokens as part of its architecture.
> >
> > The method is presented specifically as XAI for VQGMs, and the concern shared by other reviewers is that clear examples of applications are insufficiently motivated in the current revision. A point can then be made, that if the method itself has broader applications, then highlighting these by showing applications beyond XAI for VQGMs would strengthen the quality of the work.
> >
> > > [W4, Q1] *(The fundamental motivation behind the work is insufficiently established)* We appreciate this feedback. Our work enhances the interpretability and transparency of generative models, aiding in debugging, bias detection, and ethical use
> >
> > This directly relates to aforementioned W3 and Q3, and is intended to be actionable. An updated manuscript with emphasis on this important point would have gone a long way to help mitigate concerns, and communicate the importance of your method. Note that other reviewers share this concern.
> >
> > ### [W5]
> > > [W5] *(The methodology is somewhat unclear)* The saliency-based method provides fine-grained, instance-specific explanations by identifying tokens crucial for a concept in a particular image...
> >
> > This reviewer has indeed read your manuscript, and feels somewhat confident on understanding the method. The issue raised concerns how the method is currently presented. The comment was intended as an actionable issue for the manuscript, noting that the review includes the comment
> >
> > > *This could be improved by highlighting the importance of input dependence / independence, discussing how these two approaches interact, and more explicitly clarifying or showing how they are combined to form a unified interpretation with concepts.*
> >
> > Only by parsing and rereading the manuscript multiple times, did the input dependence/independence become sufficiently clear. Currently, it shows up in the manuscript without much context for the reader, and by motivating this at an earlier stage, the manuscript would likely improve in clarity for a broader group of readers.
> >
> > ### [W6]
> > > [W6] While diffusion models are indeed powerful, VQGMs offer advantages...
> >
> > Granted. We agree that work on VQGMs remains important, regardless of the state of affairs in diffusion modelling.
> >
> > ## Summary
> >
> > This years reviews are intended to provide more actionable feedback for each submission. In this case, no revisions have resulted from the reviews, and the rebuttal mostly consists of counterarguments. While these arguments may or may not be convincing in their own right, it seems like none of the reviewers feedback has lead to direct actions to improve the manuscript.
> >
> > Overall, this reviewer still finds the work somewhat unconvincing. In particular;
> > - The overall usefulness or applications of the general explanations are insufficiently motivated in the manuscript. Note that other reviewers share this concern.
> > - The saliency method applied (Simonyan et al., 2013) has been shown to be non-robust.
> > - The use of synthetic data, while chosen as a matter of convenience, is a matter of concern shared by other reviewers. The authors claim to have preliminary experiments that shows that the method works equally well with real data, but it is not clear that this will be included in the final manuscript.
> >
> > This reviewer believes the work has merit as a concept based codebook approach, but ultimately, still find the communication and motivation as an XAI method for VQGMs unsatisfactory. I find it difficult to justify a higher score for the work from the rebuttal.
> >
> > I wish the authors good luck in their continued work on the manuscript.

---

### Official Review · Reviewer_SJ24 · 2024-11-04

**Soundness:** 4
**Presentation:** 3
**Contribution:** 3
**Rating:** 6
**Confidence:** 2

**Summary:**

This paper discusses Vector-Quantized Generative Models (VQGMs) as effective tools for image generation, highlighting a gap in understanding the specific function of discrete tokens within these models' codebooks. To address this, the paper introduces Concept-Oriented Token Explanation (CORTEX), a novel methodology for interpreting VQGMs by identifying critical token combinations linked to specific concepts. CORTEX utilizes two main approaches: a saliency-based method that evaluates token significance in individual images, and an optimization-based method that searches the entire codebook to identify globally relevant tokens. The experimental results indicate that CORTEX provides clearer explanations of token usage in the image generation process compared to existing baselines and enhances the transparency of VQGMs. This improvement facilitates advanced applications such as targeted image editing and offers valuable insights into the internal representations of the models.

**Strengths:**

- The introduction of CORTEX provides a fresh perspective on interpreting VQGMs by focusing on concept-specific token combinations. This approach enhances the understanding of how models generate specific concepts, which is crucial for improving model transparency and usability.

- By addressing the challenge of distinguishing between essential and non-essential tokens, the paper contributes to the interpretability of VQGMs. This focus is particularly important in applications where understanding model decisions is critical.

**Weaknesses:**

- The saliency-based method, while effective for identifying relevant tokens, may risk overfitting to specific instances of a concept rather than providing a more generalized understanding. This could result in explanations that are too tailored to particular images, reducing their utility for broader applications

**Questions:**

- Can you provide more details on how CORTEX compares to existing explainability methods for generative models?

- How do you address the potential issue of biases present in the pretrained VQGMs affecting the explanations generated by CORTEX? What steps can be taken to ensure that the explanations are fair and unbiased?

---

> ### Author Response · Authors · 2024-11-27
> **Thanks you!**
>
> **[W1]** *The saliency-based method may risk overfitting to specific instances, reducing utility for broader applications.*
>
> **Response:** While the saliency-based method focuses on individual images, we mitigate overfitting by aggregating results across multiple images to identify common patterns and tokens that are consistently important for a concept. This aggregation helps generalize the explanations beyond specific instances.
>
> Moreover, our optimization-based method complements this by providing a global, input-independent perspective, identifying tokens that are generally important for a concept across the entire codebook. Together, these methods offer both specific and general insights, enhancing the overall utility of our approach.
>
> **[Q1]** *Can you provide more details on how CORTEX compares to existing explainability methods for generative models?*
>
> **Answer:** Current generative models primarily fall into two categories: the mainstream diffusion-based models and another widely adopted approach that uses pre-trained codebooks combined with transformers for next-token prediction to generate images or videos [1, 2, 3, 4, 5, 6, 7]. The latter, which leverages vector quantization and next-token prediction from a codebook, offers advantages such as faster generation and stronger continuity between tokens, making it one of the reasons for the growing popularity of VQGMs.
>
> Our method, CORTEX, is specifically designed for generative models utilizing codebooks, such as VQGMs. For generative models that do not use codebooks, like Stable Diffusion, our approach is not applicable. Importantly, CORTEX is the first method to propose interpreting the tokens within the codebook of a VQGM. In contrast, existing explainability methods [8, 9, 10] typically focus on pixel-level explanations rather than the token-level interpretability we aim to achieve. This distinction highlights the novelty of our approach and its specific applicability to token-based generative models.
>
> **[Q2]** *How do you address potential biases in the pretrained VQGMs affecting the explanations? What steps can be taken to ensure fairness and unbiased explanations?*
>
> **Answer:** We can address biases in pretrained VQGMs by treating bias-related concepts, such as skin tone or race, as the target concepts for explanation within our framework. By using our methods, we can identify the tokens associated with these bias-related concepts and uncover unintended correlations arising from biased training data.
>
> Our methods can help mitigate these biases in several ways:
>
> 1. **Identifying and Adjusting Tokens**     - By revealing which tokens are linked to biased concepts, we can adjust the codebook or modify the token selection process during generation.     - For instance, if certain tokens disproportionately influence the generation of sensitive concepts, we can modify, reweight, or remove these tokens to reduce their impact.
> 2.  **Rebalancing Training Data**     - Retraining or fine-tuning the VQGM with more balanced and representative data can help address underlying biases in the model.
>
> [1] Esser, Patrick, Robin Rombach, and Bjorn Ommer. "Taming transformers for high-resolution image synthesis." *Proceedings of the IEEE/CVF conference on computer vision and pattern recognition*. 2021.
>
> [2] Wang, Xinlong, et al. "Emu3: Next-token prediction is all you need." *arXiv preprint arXiv:2409.18869* (2024).
>
> [3] Yu, Lijun, et al. "Magvit: Masked generative video transformer." *Proceedings of the IEEE/CVF Conference on Computer Vision and Pattern Recognition*. 2023.
>
> [4] Yu, Lijun, et al. "Language Model Beats Diffusion--Tokenizer is Key to Visual Generation." *arXiv preprint arXiv:2310.05737* (2023).
>
> [5] Kondratyuk, Dan, et al. "Videopoet: A large language model for zero-shot video generation." *arXiv preprint arXiv:2312.14125* (2023).
>
> [6] Jin, Yang, et al. "Unified language-vision pretraining with dynamic discrete visual tokenization." *arXiv preprint arXiv:2309.04669* (2023).
>
> [7] Jin, Yang, et al. "Video-lavit: Unified video-language pre-training with decoupled visual-motional tokenization." *arXiv preprint arXiv:2402.03161* (2024).
>
> [8] Simonyan, Karen. "Deep inside convolutional networks: Visualising image classification models and saliency maps." *arXiv preprint arXiv:1312.6034* (2013).
>
> [9] Selvaraju, Ramprasaath R., et al. "Grad-cam: Visual explanations from deep networks via gradient-based localization." *Proceedings of the IEEE international conference on computer vision*. 2017.
>
> [10] Nguyen, Anh, Jason Yosinski, and Jeff Clune. "Multifaceted feature visualization: Uncovering the different types of features learned by each neuron in deep neural networks." *arXiv preprint arXiv:1602.03616* (2016).

---

### Official Review · Reviewer_EibW · 2024-11-04

**Soundness:** 2
**Presentation:** 3
**Contribution:** 2
**Rating:** 3
**Confidence:** 4

**Summary:**

The paper introduces CORTEX, a novel framework designed to enhance interpretability in Vector-Quantized Generative Models (VQGMs) by identifying concept-specific token combinations within the model’s codebook. CORTEX leverages two main methods: a saliency-based approach for analyzing token importance within individual images and an optimization-based approach to identify globally relevant tokens across the codebook. By doing so, CORTEX facilitates concept-specific explanations in VQGMs, enabling practical applications like targeted image editing. Experiments demonstrate the framework’s efficacy in interpreting token contributions, providing insights into the VQGM’s internal workings.

**Strengths:**

* The tackled explainability of vector-quantized tokens used in image generative models is an interesting problem worth exploring.
* The two proposed methods are solid defined and well presented

**Weaknesses:**

* The comparison between a CNN-based extractor and a Resnet-based extractor does not provide valuable information on the effectiveness of the proposed approach.
* It remains unclear how useful the explanation obtained via both methods potentially is. Possibilities for identifying biases, improving robustness, etc. need to be clarified to justify the proposed design.
* The experiments are limited on imagenet with class-conditioned generation, whereas the more general text to image setup where explanation might be more important is not explored.

**Questions:**

* Since the submission focuses on the explanation of the tokens used during generation, can some visualization of the token space rather than pixel space be provided as a better presentation of the explanation process?
* Although image tokens are usually strongly correlated to the corresponding pixel region, simply masking the pixel region remains questionable as the token could have affected other pixels during the token-to-pixel decoding. Have the authors trying masking the tokens before decoding to pixels?
* What would be a reasonable setup to evaluate the proposed methods on text-to-image models?
* How to present some empirical and quantitative results to justify the usefulness of the explanations produced by the proposed methods? What new capabilities could be possibly enabled if we can explain them?

---

> ### Author Response · Authors · 2024-11-27
> **Thank you! (1/2)**
>
> **[W1]** *The comparison between CNN-based and ResNet-based extractors may not provide valuable insights into the effectiveness of the proposed approach.*
>
> **Response:** Our approach to training the information extractor is designed to ensure it learns token-level features, essentially functioning as a "knowledge storage unit." It is expected that different extractor architectures will capture different patterns, which is entirely natural. However, our goal is not to explain the information extractor itself but rather to leverage the knowledge it encodes to provide human-understandable interpretations of high-dimensional tokens.
>
> Importantly, **the extractor is not fixed; it is trained specifically for the concepts that the user is interested in**. For example, if the concept involves positional information, the extractor architecture should be sensitive to spatial features. Our experiments are not intended to demonstrate that a CNN-based extractor can capture all token information. Instead, they aim to validate the effectiveness of our method: that even with a simple extractor architecture, we can interpret the tokens in VQGM, which are otherwise difficult for humans to understand.
>
> Our proposed method and experiments are designed to demonstrate that our extractor can learn token-based information and transform it into human-understandable explanations. The goal is not to prove that our extractor has learned all possible information. The choice of information extractor depends on the specific concept being targeted, as different concepts may require different extractors.
>
> **[W2]** *The usefulness of the explanations obtained is unclear. Clarification on identifying biases, improving robustness, etc., is needed to justify the design.*
>
> **Response:** We want to clarify that our explanations are useful in applications including image editing, bias detection, etc.
> -As for the optimization-based method, as shown in Figure 4 of the paper, the explanations generated can be directly applied to image editing. By leveraging the results of the optimization process, specific aspects of an image can be controlled, altered, or refined, showcasing the practical utility of this approach.
> -The saliency-based method provides explanations for the token inputs of already generated images. In cases where the concept being interpreted is undesirable, such as bias (e.g., related to skin tone or race), the identified tokens can be edited, by deleting or modifying the corresponding tokens in the codebook, and the VQGM can be re-trained to remove these biases.
>
> **[W3]** *Experiments are limited to ImageNet with class-conditioned generation. The text-to-image setup is not explored, where explanation might be more important.*
>
> **Response:** We want to clarify that our proposed explanation method is applicable to any VQGM with a discrete codebook, regardless of the number of concepts or the size of the codebook.
> We demonstrate that with a small external model (the information extractor), we can generate human-understandable interpretations for VQGM tokens.
> This interpretability approach can be applied to any VQGM model that utilizes a codebook.
> Moreover, our concepts can be defined as the context for the model's prompt in a text-to-image model. By training a simple information extractor tailored to the concept that needs to be interpreted, token-based concept explanations can be obtained. Therefore, our method is broadly applicable to any model utilizing a codebook and can provide interpretations for any concept.

---

> ### Author Response · Authors · 2024-11-27
> **Thank you! (2/2)**
>
> **[Q1]** *Can visualization of the token space be provided for better presentation of the explanation process?*
>
> **Answer:** Thank you for your suggestion. The reason we did not use token space visualization is that our explanation focuses on making tokens understandable from a human visual perspective, such as identifying which tokens correspond to specific features. Our method aims to explain the relationship between tokens and concepts rather than clustering or analyzing the relationships within the high-dimensional token space. Therefore, our approach emphasizes providing explanations that are visually interpretable for humans rather than visualizing the token space itself.
>
> **[Q2]** *Have the authors tried masking the tokens before decoding to pixels?*
>
> **Answer:** Yes, we have considered masking the tokens before decoding. Removing a token would corrupt its corresponding spatial region in the decoded image, replacing it with visual artifacts. This introduces additional uncertainty and does not differ significantly from simply masking the corresponding token region in the decoded image. Since each token corresponds to a fixed region in the image, directly masking the decoded region provides the same effect while being more controlled and consistent. Therefore, we chose to mask the token regions in the decoded images with black to achieve a more interpretable and reasonable comparison.
>
> **[Q3]** *What would be a reasonable setup to evaluate the proposed methods on text-to-image models?*
>
> **Answer:** As mentioned in **[W3]**, our method is applicable to any VQGM that utilizes a codebook, including text-to-image models. Following the approach described in our paper, the text prompt can be treated as the concept to be explained. By training a corresponding information extractor for this concept, the extractor can identify the token combinations associated with the given concept. Thus, our framework can be seamlessly applied to any text-to-image model and can provide explanations for any concept of interest.
>
> **[Q4]:** *How to present empirical and quantitative results to justify the usefulness of the explanations? What new capabilities could be enabled?*
>
> **Answer:** As mentioned in **[W2]**, our saliency-based method can identify tokens in already generated images that correspond to concepts related to biases. By editing these specific tokens in the codebook, we can mitigate or eliminate such biases. Additionally, our optimization-based method enables image editing by optimizing token combinations to achieve desired modifications. These capabilities demonstrate the practical usefulness of our explanations, enabling both bias mitigation and fine-grained control over generated images.

---

### Official Review · Reviewer_cJrD · 2024-11-04

**Soundness:** 3
**Presentation:** 3
**Contribution:** 2
**Rating:** 5
**Confidence:** 4

**Summary:**

This paper proposes a novel approach, named CORTEX, for interpreting token selections under given specific concept-related conditions in VQGMs. The authors develop a framework based on information extraction and derive two methods, i.e., saliency-based method and optimisation-based method, to analyse token combination modes for case-dependent and case-independent scenarios. Extensive experiments conducted under various settings indicate these two methods can well explain the mode of token combinations in VQGMs, thus demonstrating the effectiveness of proposed method and framework.

**Strengths:**

- There is a lack of sufficient interpretability and controllability analysis in the field of VQGM. This paper provides an effective supplement to this field.
- The author's writing is relatively clear and fluent.
- The author(s) conduct various experiments to comprehensively evaluate the quality of the proposed model from both qualitative and quantitative perspectives.

**Weaknesses:**

- Lack of some context, such as the source and performance of the VQGAN used.
- The discussed domain is relatively narrow. It can be better if there are further discussions about the generalisation of proposed methods in a broader context, such as discussing what kind of effects can be bring in when the number of different concepts and the size of codebook vary, or how these methods can be applied in other tasks (e.g., text-to-image) .
- It would be better if the author(s) could further analyze the relationship between saliency-based and optimization-based methods, and under what circumstances which one can provide more information / observations, rather than simply discussing the two methods side by side based on whether they are related to input.

**Questions:**

- Regarding saliency-based method, the author(s) estimate TSV based on the absolute value of the gradient of $E$. Is it possible that the values in various dimensions of the embedding itself will affect the result? What if all the code entries have been normalized into unit vectors?

- Regarding information extractors, it seems that there is tiny performance difference between these two extractors in Figure 3, Table 2 and Table 3. The author(s) attribute this to the insensitivity to extractor structure. However, RE and CE are actually both CNN-based extractors. Will the author(s) consider other kinds of structures like transformers?

- Regarding the optimization-based method, the author(s) choose concept pairs with high similarity, and add a mask to study how tokens evolve in a small region. What would happen if we choose two concepts with a large distance in semantics and / or remove the mask constraint?

- Regarding the pretrained model, the sensitivity of ViT and CNN is different. Is there related analyses to this observation?

- The author(s) mention that the proposed methods can help improve controllability of VQGMs. How can these two methods be used to improve the controllability of VQGMs?

---

> ### Author Response · Authors · 2024-11-27
> **Thank you ! (1/2)**
>
> **[W1]** *Lack of context about the source and performance of the VQGAN used.*
>
> **Response:** We used the pretrained VQGAN model [1], trained on ImageNet. This model achieves competitive performance in image generation tasks, effectively capturing complex visual features. The performance of the VQGAN is a well-recognized conditional image generation model, but it is out of the scope of our paper. Therefore, we did not add it to our discussion. The detailed performance report of VQGAN can be found in [1].
>
>
> **[W2]** *The domain is relatively narrow. Further discussions on generalization, effects of varying concepts and codebook size, and applications to other tasks like text-to-image would be beneficial.*
>
> **Response:**
> First, we want to clarify that our approach is applicable to any VQGM with a discrete codebook, regardless of the number of concepts or the size of the codebook.
>
> Regarding your concern about the domain being relatively narrow, it’s worth noting that current generative models primarily fall into two categories: diffusion or vector-quantized transformers. While diffusion models denoise random noise iteratively, VQ-transformers encode images into discrete tokens for autoregressive generation, offering faster inference and better semantic coherence [1, 2, 3, 4, 5, 6, 7].
>
> This interpretability approach can be applied to any VQGM model that utilizes a codebook.
> Moreover, our concepts can be defined as the context for the model's prompt in a text-to-image model. By training a simple information extractor tailored to the concept that needs to be interpreted, token-based concept explanations can be obtained. Therefore, our method is broadly applicable to any model utilizing a codebook and can provide interpretations for any concept.
>
> To the best of our knowledge, our method is the first to interpret tokens in the codebook. We demonstrate that with a small external model (the information extractor), we can generate human-understandable interpretations for VQGM tokens.
>
> **[W3]** *Further analysis of the relationship between saliency-based and optimization-based methods is needed, rather than discussing them side by side.*
>
> **Response:** Thank you for this suggestion. The saliency-based method provides fine-grained, instance-specific explanations by identifying tokens crucial for a concept in a particular image. In contrast, the optimization-based method offers a global perspective by finding token combinations representing a concept across all possible images, independent of specific inputs.
>
> Under circumstances where an explanation is needed for a specific generated image, the saliency-based method is more informative. When seeking a general understanding of how a concept is represented in the model's codebook, the optimization-based method is more appropriate.
>
> Therefore, the saliency-based method interprets tokens at the level of specific generated images, while the optimization-based method explains the entire codebook. Additionally, the results of the optimization-based method can also be used for tasks such as image editing.
>
> We will revise our paper and add this discussion to the final version.
>
> [1] Esser, Patrick, Robin Rombach, and Bjorn Ommer. "Taming transformers for high-resolution image synthesis." *Proceedings of the IEEE/CVF conference on computer vision and pattern recognition*. 2021.
>
> [2] Wang, Xinlong, et al. "Emu3: Next-token prediction is all you need." *arXiv preprint arXiv:2409.18869* (2024).
>
> [3] Yu, Lijun, et al. "Magvit: Masked generative video transformer." *Proceedings of the IEEE/CVF Conference on Computer Vision and Pattern Recognition*. 2023.
>
> [4] Yu, Lijun, et al. "Language Model Beats Diffusion--Tokenizer is Key to Visual Generation." *arXiv preprint arXiv:2310.05737* (2023).
>
> [5] Kondratyuk, Dan, et al. "Videopoet: A large language model for zero-shot video generation." *arXiv preprint arXiv:2312.14125* (2023).
>
> [6] Jin, Yang, et al. "Unified language-vision pretraining with dynamic discrete visual tokenization." *arXiv preprint arXiv:2309.04669* (2023).
>
> [7] Jin, Yang, et al. "Video-lavit: Unified video-language pre-training with decoupled visual-motional tokenization." *arXiv preprint arXiv:2402.03161* (2024).

---

> ### Author Response · Authors · 2024-11-27
> **Thank you ! (2/2)**
>
> **[Q1]** *Could the values in various dimensions of the embedding affect the TSV results? What if all code entries are normalized into unit vectors?*
>
> **Answer:**  The TSV values are calculated based on the gradient of the output with respect to each input token, and these tokens are directly drawn from the codebook. During pretraining, the tokens in the codebook are optimized based on Euclidean distance. The main goal of our study is to interpret the pre-trained tokens in VQGM codebook. Normalizing the embeddings into unit vectors would alter the original information encoded in the tokens, making it impossible to reconstruct the original image using the VQGM decoder faithfully. This is why we cannot use unit vectors to calculate TSV.
>
> **[Q2]** *Will the authors consider other extractor architectures like transformers?*
>
> **Answer:**  Our approach to training the information extractor is designed to ensure it learns token-level features, essentially functioning as a "knowledge storage unit." It is expected that different extractor architectures will capture different patterns, which is entirely natural. However, our goal is not to explain the information extractor itself but rather to leverage the knowledge it encodes to provide human-understandable interpretations of high-dimensional tokens.
>
> Importantly, **the extractor is not fixed; it is trained specifically for the concepts that the user is interested in**.
> Our experiments are not intended to demonstrate that a CNN-based extractor can capture all token information. Instead, they aim to validate the effectiveness of our method: that **even with a simple extractor architecture, we can interpret the tokens in VQGM**.
>
>
> **[Q3]** *What happens if we choose two semantically distant concepts or remove the mask constraint in the optimization-based method?*
>
> **Answer:**  Thank you for your insightful suggestion. The reason we focus on semantically similar concepts in our experiments is to demonstrate that **our information extractor can learn fine-grained, token-level distinctions**. For example, when optimizing for one bird species, we show that modifying a specific region can lead the pre-trained model to classify the image as another bird species. This highlights that our model has effectively captured the unique characteristics of each bird species, further validating the feasibility of our method.
>
> As for removing the mask constraint, it is certainly possible. However, in this case, as illustrated in Figure 4 with our image editing example, the entire image will turn into noise at the beginning of optimization, making it unrelated to the original input. The purpose of this experiment is to show that by optimizing only a specific region, we can increase the pre-trained model's prediction confidence for the target label. This demonstrates that our information extractor successfully captures fine-grained differences between concepts.
>
> **[Q4]** *Is there related analysis on the different sensitivities of ViT and CNN models?*
>
> **Answer:** As mentioned in Q2, our experiments are not designed to prove that the information extractor can learn all token information, nor are they aimed at interpreting the information extractor itself. Instead, we demonstrate that even a simple CNN-based information extractor can effectively capture concept-related information stored in tokens.
>
> Additionally, our information extractor is not fixed; it is trained specifically for the concept of interest to the user. Therefore, the primary goal of our experiments is to validate that the proposed information extractor can learn meaningful information from the tokens and that our interpretability approach is feasible and effective.
>
> **[Q5]** *How can these methods improve the controllability of VQGMs?*
>
> **Answer:**  By identifying tokens critical for specific concepts, our methods provide different utilities in the context of VQGMs. The optimization-based method can be directly applied to image editing by modifying token combinations to control the presence or absence of certain features in generated images. For example, as demonstrated in Figure 4, the optimization-based method allows targeted edits, such as changing the species of a bird or altering the style of an object, enabling fine-grained customization and enhancing controllability in the generation process.
>
> In contrast, the saliency-based method identifies tokens within existing images that are associated with specific concepts. This can be used for tasks like detecting and addressing biases in the model. For instance, if tokens related to certain biases are identified, they can be removed from the codebook, and the model can be re-trained to mitigate those biases, improving fairness and reducing unwanted artifacts in the generated outputs.

---

### Meta-Review · Area_Chair_tLAA · 2024-12-17

**Metareview:**

This paper introduces a vector quantized (VQ) method and proposes two novel techniques: a saliency-based approach and an optimization method that embeds semantic meaning into tokens, enhancing the interpretability of VQ models.
Strengths: The reviewers acknowledge the effectiveness of incorporating a VQ model and the novelty of the proposed methods, addressing a gap in current literature. This sentiment is echoed by several reviewers. The proposed techniques are verified on a targeted editing application, demonstrating its applicability.

Weaknesses:
* W1 The writing lacks clarity, and the overall methodology is not well-contextualized.
* W2 Clarification is needed regarding the proposed techniques, particularly the effect of the two methods on interpretability.
* W3 The saliency detection method relies on a paper from 2013, which has known limitations that may impact its ability to capture interpretive information.
* W4 The empirical validation is lacking. The current verification process is limited to a closed vocabulary from ImageNet, which contains simple object categories and may not reflect broader, real-world challenges. Explanations related to interpretability, are not demonstrated for the understanding task such as identifying bias or improving robustness. The use of a synthetic dataset raises concerns about whether the results generalize sufficiently.

Unfortunately, there was no discussion between the reviewers and the authors. In the end, three reviewers voted to reject the paper; and only one leaned towards acceptance, but with low confidence. Given the rating, and that the AC cannot fully confirm that the authors' responses have adequately addressed the clarifications regarding the technical questions., the AC recommends not accepting this submission.

**Additional Comments On Reviewer Discussion:**

There was no discussion between the reviewers and the authors. The Area Chair reviewed the authors' response to the major weaknesses listed above. While many of the weaknesses appear to be addressed, It seems to the AC that W4 may not be fully addressed. In the rebuttal, the authors pointed out the value of their existing experiments but did not provide further verification to address the concern. The verification could be made more substantial by including a more challenging dataset with multiple objects and open vocabulary. Additionally, the evaluation of semantic tokens could be improved by demonstrating their connection to understanding tasks.

---

### Decision · Program_Chairs · 2025-01-22

Reject